# Acute Changes in Myocardial Work during Isometric Exercise in Hypertensive Patients with Ischemic Heart Disease: A Case–Control Study

**DOI:** 10.3390/jcm13195955

**Published:** 2024-10-07

**Authors:** Giuseppe Caminiti, Maurizio Volterrani, Ferdinando Iellamo, Giuseppe Marazzi, Valentino D’Antoni, Camilla Calandri, Sara Vadalà, Matteo Catena, Deborah Di Biasio, Vincenzo Manzi, Valentina Morsella, Marco Alfonso Perrone

**Affiliations:** 1Department of Human Science and Promotion of Quality of Life, San Raffaele Open University, 00163 Rome, Italy; giuseppe.caminiti@uniroma5.it; 2Cardiology Rehabilitation Unit, IRCCS San Raffaele Roma, 00166 Rome, Italyvalentino.dantoni@sanraffaele.it (V.D.);vada.sara21@gmail.com (S.V.); matteo.catena94@gmail.com (M.C.); debbydibiasio20@gmail.com (D.D.B.); valentina.morsella@sanraffaele.it (V.M.); 3Division of Cardiology and Sports Medicine, Department of Clinical Sciences and Translational Medicine, University of Rome Tor Vergata, 00133 Rome, Italy; iellamo@uniroma2.it (F.I.); calandricamilla@gmail.com (C.C.); marco.perrone@uniroma2.it (M.A.P.); 4Department of Wellbeing, Nutrition and Sport, Pegaso Open University, 80132 Naples, Italy; vincenzo.manzi@unipegaso.it

**Keywords:** isometric exercise, ischemic heart disease, myocardial work

## Abstract

**Background:** The acute hemodynamic response to isometric exercise in hypertensive patients’ ischemic heart disease (IHD) has been poorly investigated. The aim of this study was to assess acute changes in left ventricular myocardial work (MW) during isometric bilateral knee extension in patients with IHD. **Methods:** Twenty stable hypertensive patients with IHD and ten healthy, age-matched controls (HC) were enrolled. All subjects performed an isometric knee extension exercise at 30% of their maximal voluntary contraction. The effort was maintained for three minutes or until exhaustion. At baseline, at peak exercise, and after 10 min of recovery, echocardiography evaluation was performed and blood pressure (BP) and heart rate (HR) were measured. **Results:** The exercise was well tolerated by all subjects. At peak exercise, systolic BP in the IHD was significantly higher than HC (37.6 ± 7.2 vs. 8.4 ± 2.3 mmHg; *p* 0.002). The HC group had a greater increase in HR than IHD (19.7 ± 6.2 vs. 8.4 ± 2.2 bpm; *p* 0.009). The E/E′ ratio increased in IHD and was unchanged in the control group. The global work index increased significantly in IHD compared to HC (+15% vs. +3%; *p* 0.026). Global constructive work increased significantly in IHD compared to HC (+29.8% vs. +7.4 respectively, *p* 0.031). Global wasted work increased by 92.3% in IHD and was unchanged in HC. The global work efficiency decreased in IHD (−18%), but was unchanged in HC (between-groups *p* 0.019). Stroke volume decreased in IHD and was unchanged in HC. Cardiac output was unchanged in IHD, while it increased in HC. **Conclusion:** In patients with hypertension and underlying IHD, an acute isometric load causes a great increase in systolic BP and LV filling pressure. It follows a mostly ineffective increase in MW that fails to maintain stroke volume.

## 1. Introduction

Exercise training is a well-established non-pharmacologic intervention for the prevention and treatment of hypertension [1,2]. While, traditionally, most studies have focused on the anti-hypertensive effects of continuous, moderate-intensity, aerobic exercise [3], beneficial effects elicited by resistance training or protocols combining aerobic and resistance training have also been underlined [4,5]. Resistance exercise that includes dynamic and isometric contractions appears to be particularly indicated for aged, hypertensive patients carrying chronic conditions, including those with cardiovascular diseases [2]. In these patients, resistance exercise has several beneficial effects beyond the reduction in blood pressure (BP), including increased skeletal muscle trophism and strength, improvement of body composition, and prevention of skeletal muscle loss. These effects help patients to preserve their ability to perform daily activities and, ultimately, contribute to improving their quality of life [6,7,8,9]. Isometric exercise (IE), characterized by sustained muscular contractions in which the length of the muscle does not change, has emerged as a convenient, time-efficient intervention whose anti-hypertensive effects have been supported in multiple meta-analyses and appear to be similar to or greater than those observed in traditional aerobic exercise training [5,10,11]. However, IE has been less extensively utilized than dynamic resistance exercise in the cardiac prevention and rehabilitation fields, the major concern being the fear of incurring an excessive rise in BP, with a correspondent unwanted increase in cardiac work. This aspect seems to be particularly relevant for hypertensive patients with underlying ischemic heart disease (IHD), since maintaining a low double product during exercise is a primary therapeutic goal in this context. However, the actual cardiac load occurring during IE is variable, since the rise in BP depends on the mass of the muscle involved as well as on the intensity of the effort. It has been shown that, when using small-muscle mass, isometric and dynamic resistance exercises evoked equal increases in BP [12]. Some authors have encouraged handgrip IE prescription in hypertensive IHD patients because they have observed adequate hemodynamic responses [13,14]. Conversely, the hemodynamic response to bilateral isometric knee extension in hypertensive patients with IHD has been less extensively studied.

### Study Hypothesis

Pursuing the idea of implementing the use of IE for counteracting hypertension in patients undergoing secondary prevention/rehabilitation programs, it is mandatory to preliminarily assess the tolerability of IE in IHD patients. This is intended to investigate how performing IE acutely affects cardiac function and contractile efficiency in patients evaluated before entering cardiac rehabilitation programs. A non-invasive detailed reconstruction of myocardial function is possible today through speckle tracking echocardiography and, in particular, through myocardial work (MW), which has been introduced in recent years as a non-invasive measurement of left ventricular (LV) function. MW uses a combination of LV global longitudinal strain (GLS) and systolic BP to obtain LV pressure–strain loops [15,16]. It provides valuable information regarding myocardial contraction efficiency that is particularly relevant in the interpretation of myocardial performance during exercise [17]. In the present study, we investigate changes in myocardial efficiency (MWE) as part of the acute hemodynamic response elicited by bilateral isometric knee extension exercise in untrained hypertensive patients with IHD and compare them with changes occurring in healthy subjects performing the same exercise.

## 2. Materials and Methods

Population: The study included a total of 30 subjects. Twenty of them were patients with diagnosed IHD, and ten were healthy age-matched controls. Patients with IHD were referred to the outpatient’s service of cardiac rehabilitation of San Raffaele IRCCS of Rome. The following inclusion criteria were adopted: age over 45 years and previous diagnosis (made at least 6 months before recruitment) of arterial hypertension and of IHD, with the latter including: history of stable angina; unstable angina/myocardial infarction; and previous percutaneous coronary revascularization or previous coronary artery bypass grafting. Patients had to be in optimal medical therapy, in stable clinical condition, and with a stable sinus rhythm for at least three months. The following exclusion criteria were adopted: myocardial ischemia or threatening arrhythmias during the resting assessment and/or during the ergometric test; previous diagnosis of heart failure; permanent atrial fibrillation; baseline blood pressure levels at rest over 160/100 mmHg; anaemia with haemoglobin levels below 11 g/dL; concomitant diagnosis of chronic respiratory disease with documented FEV1 below <50%; previous diagnosis peripheral artery disease with exercise-limiting claudication; and poor acoustic window. Subjects with concomitant severe heart valve diseases, those with a previous diagnosis of hypertrophic cardiomyopathy, subjects being recently (less than six months) engaged in exercise training programs, and those who declared to practice spontaneous regular exercise (more than 2 sessions/week) were also excluded. Healthy controls were found among relatives of the patients enrolled and among relatives of the hospital staff. They did not have to take hemodynamic drugs and did not have to practice regular sports. The study complied with the Declaration of Helsinki and was approved by the local Ethics Committee of San Raffaele IRCCS (protocol number 18/2023, approval date: 21 March 2023). All patients gave written informed consent before entering the study.

Study design: The study flow chart is summarized in Figure 1. All participants were screened in a preliminary visit. During this visit, clinical history and anthropometric data, including body mass index (BMI), resting heart rate (HR), and BP, were collected. Subjects meeting the inclusion and exclusion criteria were then invited to join the study, and those who agreed and signed up the informed consent were then reconvened for a second visit that was scheduled within a week from the first one. During the second visit, patients with IHD performed a symptom-limited ergometric test on a bike (Mortara Instrument, Casalecchio Di Reno, Bologna, Italy) to rule out ischemic or arrhythmic conditions. In the same day, all subjects were familiarized with the dynamometer and completed a double-leg isometric extension exercise test. For each subject, the experimental session was scheduled within another week from the second visit. Subjects of the HC group were tested on a non-working day.

Experimental sessions were held in the gym of the rehabilitation facility of S. Raffaele IRCCS. All tests were performed on a knee flex/extension dynamometer (Technogym Wellness System, Technogym, Cesena, Italy), and participants were asked to perform isometric double-leg (knee extension) exercises. All participants were asked to avoid performing strenuous exercise within 24 h and agreed to abstain from alcohol and caffeine for a 12-h period prior to the experimental session. Participants were seated on the dynamometer, and the seat was adjusted appropriately for each individual so that the axis of rotation around the dynamometer shaft was adjacent to the lateral femoral condoyle of the subject’s right leg. Subjects had their knees bent at a 90° angle. Both legs were positioned underneath the knee extension/flexion attachment arm of the dynamometer. Participants positioned their arms along their trunk so that the left arm did not obstruct the sonographer responsible for acquiring the echocardiographic data, who was positioned on the left side of the examined subject. A manual sphygmomanometer cuff was placed on the right arm. The test for determining maximal voluntary contraction (MVC) consisted of 3 maximal contractions, each one lasting 3–5 s each, with 1 min of rest between contractions [18]. During each contraction, subjects were encouraged by the evaluator to exert their maximal effort. The maximum force generated by the patient was recorded for each trial, and the maximum value over the three trials was used as the final measurement. After the test, subjects were asked to rest for 30 min, at the end of which they performed IE: the intensity of the exercise was set at 30% of their MVC and the duration of the exercise was 3 min. During the effort, subjects were invited to breathe regularly in order to avoid the Valsalva manoeuvre. Echocardiographic assessments and BP measurements were made at rest, during the exercise, and after 10 min of recovery. The ultrasound acquisition during the exercise phase started 90 s from the beginning of the exercise, since we estimated that it would take about one minute to carry out all the required assessments.

Transthoracic echocardiography: The echocardiography was performed with patients in the sitting position with the sonographer placed on the left side of the subject being examined. A cardiovascular ultrasound Vivid E95^®^ (GE Healthcare, Chicago, IL, USA) with a 4.0-MHz transducer was used for the echocardiography examinations during the entire study. Imaging windows and measurements were obtained according to the current guidelines of the European Association of Cardiovascular Imaging [19]. During each examination, one-lead electrocardiography monitoring was positioned on the chest of the subject. After the imaging acquisition, all the echocardiographic images were digitally stored, and their analysis was performed offline. During the review process, an experienced technician performed deformation measures by using proprietary software (version 10.8, EchoPAC; GE Vingmed Ultrasound, Horten, Norway). Measures of left ventricular end-diastolic volume (LVEDV) and left ventricular end-systolic volume (LVESV) were obtained from the apical two and four chamber windows; LVEF was then calculated by using the modified Simpson’s method. Stroke volume (SV)was then calculated as EDV − ESV, cardiac output (CO) as HR × SV, and the ejection fraction (EF) as EF = (EDV − ESV)/EDV). Measures of left atrium (LA) volume were obtained from apical four-chamber and three-chamber views at end of the systole, before the opening of the mitral valve, and before the biplane Simpson’s method of disks was adopted. The LA volume index (LAVI) was obtained by dividing the LA volume by the body surface area of the study subjects. The E/A ratio was defined as the ratio between the E wave, corresponding to the peak left ventricle filling velocity in early diastole, and the A wave corresponding to peak velocity flow in late diastole. Colour tissue Doppler tracings were performed in the 4-chamber view; the range gate was placed at the lateral mitral annular segments. The E/E′ ratio was defined as the ratio between E wave velocity and the average between septal and lateral LV E′ wave velocities. Measurements for LV global longitudinal strain (LVGLS) were obtained from through two-, three-, and four-chamber views. Despite the software automatically detecting LV endocardial boundaries, whenever deemed appropriate, the images were edited in order to conform to the visualized LV boundaries. The software measured the maximum negative value of strain during the systole, and this value was considered as the maximum contractility for each segment. We calculated LVGLS by considering the average values from each segment. LA strain was assessed through two- and four-chamber views. Measures of LA deformation tracking were made by using the R-R gating, in which the R wave represented the starting point. The software automatically traced the endocardial and epicardial contours of the LA. Again, if necessary, manual adjustments were made. A set of control points was automatically placed on the middle curve of the myocardial wall in the reference phase based on the drawn endocardial and epicardial contours. Longitudinal strain curves were generated for each segment by the software program, and an average curve of each segment was calculated. The LA reservoir strain, conduit strain, and contractile strain were obtained from the subdivision of longitudinal strain measurements. PALS was defined as a positive peak during LV systole at the end of the atrial diastolic phase; PACS was defined as a positive peak during early LV diastole, before the start of the atrial systolic phase. MW was evaluated from mitral valve closure to mitral valve opening. A 17-segment bull’s eye with the segmental and global work index (GWI) corresponding to the area within the curve total work from mitral valve closure to mitral valve opening was obtained. Global constructive work (GCW) was defined as the work performed during shortening in systole and the negative work performed during lengthening in isovolumetric relaxation. Global wasted work (GWW) corresponded to the negative work performed during lengthening in systole plus the work performed during shortening in isovolumetric relaxation. Global work efficiency (GWE) was defined as constructive work divided by the sum of constructive and wasted work. Valvular event times were identified by using pulse-wave Doppler recordings at the mitral valve and aortic valve level. Confirmation of valvular events was performed by 2D evaluation of the apical long-axis view [15].

### Statistical Analysis

Data were expressed as mean ± SD. The assumption of normality was checked using the Shapiro–Wilk hypothesis test. Data obtained at rest, during exercise, and at recovery were compared by using repeated-measures two-way ANOVA with Bonferroni corrections for post hoc testing. The Pearson correlation coefficient test was used to measure the strength of the linear association between two variables. The level of significance was set at *p* < 0.05. The statistical program, IBM SPSS Statistics v26.0, was used for the processing, presentation, and statistical analysis of the data.

## 3. Results

The isometric exercise was well tolerated by all participants; no side effects occurred during the experimental sessions. Muscle exhaustion during the isometric effort occurred in eight IHD patients and in none of the HC group. Characteristics of the patients and controls are summarized in Table 1. Sixteen out of twenty patients of the IHD groups had had a previous myocardial infarction. Patients of the IHD groups were taking, on average, 2.3 ± 1.1 drugs for the treatment of hypertension. All of them were taking antiplatelet agents, statins, and betablockers. Four subjects of the HC had hypercholesterolemia; three of them were taking statins and one ezetimibe. At rest, values of EF, LVGLS, GWI, and GCW were significantly higher, while GWW and GWE were lower in HS compared to IHD. GWE was significantly higher in HS compared to IHD. The double product was similar between the IHD and HC groups.

At peak exercise, systolic BP presented a greater increase in the IHD group comped to HC (37.6 ± 7.2 vs.8.4 ± 2.3 mmHg; *p* 0.002); diastolic BP presented a greater increase in the IHD group compared to HC (19.4 ± 4.6 vs. 7.4 ± 2.7 mmHg; *p* 0.036). HR presented a greater increase in the HC group compared to IHD (19.7 ± 6.2 vs. 8.4 ± 2.2 bpm; *p* 0.009). Double product increased in a similar way in the two groups. LVDV and LVSV decreased significantly in the IHD, but were unchanged in the HC (−18.1 ± 4.6 and −9.3 ± 2.7 vs. 0.5 ± 0.6 and −0.8 ± 0.2 respectively; *p* 0.041 and *p* 0.038). No significant changes occurred in LVEF or in the two groups. LVGLS decreased significantly in IHD and was unchanged in HC (Table 2). Deceleration time decreased significantly, and the E/E′ ratio increased significantly compared to rest values only in the IHD group. PALS decreased significantly in IHD compared to HC. PACS at peak exercise was unchanged compared to the rest condition in both groups. GWI increased significantly in IHD compared to HC (+15% vs. +3%; *p* 0.026). GCW increased by30.4% in the IHD group and was unchanged in the HC group (+29.8% vs. +7.4 respectively, *p* 0.031). GWW increased by91.4% in the IHD group and was unchanged in the HC group (between-groups *p* 0.0001). GWE decreased by18% in the IHD group and was unchanged in HC (between-groups *p* 0.019) (Figure 2). SV decreased in IHD and was unchanged in HC. CO was unchanged in IHD, while it increased in HC (Figure 3). In the IHD group, changes in GWW were significantly related to changes in the E/E′ ratio (r = 0.45; *p* 0.003).

## 4. Discussion

### 4.1. Main Results

A growing body of literature indicates that IE training can produce BP reductions greater than those obtainable following the currently recommended exercise guidelines and similar to those of current anti-hypertensive monotherapy [20]. However, in hypertensive patients with underlying IHD, there is a substantial risk that IE can lead to a maladaptive acute response with an abnormal rise in BP and an excessive increase in myocardial oxygen consumption [16,21,22]. A preliminary assessment of the acute hemodynamic response to IE can help to identify the best-tolerated isometric protocol for these patients. In the present study, we investigated the acute hemodynamic response evoked by isometric knee extension performed at 30% of MVC in hypertensive patients with IHD and compared it with the response evoked by the same exercise in HC. We focused particularly on changes in myocardial work, which we assessed non-invasively by speckle tracking echocardiography. We observed that the response of IHD patients was characterized by a significant increase in systolic and diastolic BP, a modest increase in HR, and an increase in inotropism indicated by a significant rise in GWI. However, the MW increase model was dysfunctional, with a clear prevalence of GWW over GCW and a consequent significant reduction in GWE. The exercise-induced rise in GWW meant that the increased contractility did not translate into hemodynamically productive work; on the contrary, the significant increase in LV uncoordinated contractions determined a significant loss in myocardial cell energy, with a reduction in contraction efficiency at peak exercise [23]. This LV inefficient contraction was associated with a decrease in SV at peak exercise. The decrease in SV, together with the inadequate increase in HR, was responsible for the lack of increase in CO registered in this group at peak exercise. Conversely, in HC, at peak exercise, there were no changes in GWI or GWE; the increase in CO was driven by a significant increase in HR, and only modest changes in systolic and diastolic BP occurred in comparison to the rest condition. Interestingly, we found that, in the IHD group, resting values of GWI, GCW, and GWE were lower, while GWW was higher compared to HC. These results are consistent with previous studies showing that MW indices are impaired in patients developing LV ischemic remodelling after myocardial infarction and in those undergoing coronary revascularization regardless of the presence of heart failure [24,25]. The resting reduction in GWE has been ascribed to a chronic impairment of energy metabolism that occurs in the remodelled myocardium [26] and has been associated with lower performance during exercise in athletes [27]. Our data suggest that the use of MW indices allows us to describe the LV performance during exercise in more depth than other methods, such as GLS or EF. In particular, the assessment of MW, before starting a cardiac rehabilitation program, could help clinicians to identify combinations of loads and exercise types that are associated with less myocardial wasting work and with greater contractile efficiency. Clearly, further investigations are needed in order to confirm our results and to explore potential long-term clinical advantages of performing speckle tracking echocardiography during exercise in clinical practice.

### 4.2. Comparison with Previous Studies

The inotropic activation that we documented in IHD, together with the significant increase in systolic BP, can be ascribed to the exercise-mediated activation of the sympathetic nervous system (SNS). An abnormal activation of SNS leading to an increase in systemic vascular resistance and to an inotropic response has been widely described during IE [23,28]. However, the involvement of SNS remains speculative in the present study, since we did not measure catecholamine levels or other indices of SNS activation. The hemodynamic response to IE that we observed in IHD was also characterized by a blunted increase in HR. An inadequate increase in HR during exercise is often observed in patients with IHD, and this is related to myocardial ischemia and autonomic dysfunction [29]. However, the systematic use of betablockers may have contributed to generating or amplifying this chronotropic incompetence in our IHD patients. In experimental models where subjects with normal LV function and with fixed HR at rest values performed IE, SV increased and compensated for the lack of a chronotropic response, allowing for CO to rise and to produce the pressor response [30,31]. In such conditions, Nobrega et al. [31] demonstrated that a combination of increased contractility and the Frank–Starling mechanism was responsible for that increase in SV during IE. Conversely, in our study, in IHD patients, SV at peak exercise decreased, leaving us to hypothesize that, in pathologic conditions, these two mechanisms are not able to provide sufficient compensation. We found that, in the IHD group, LVEDV decreased at peak exercise compared to rest values; this would imply that the Frank–Starling mechanism was not used by these patients. The decrease in LVDD denotes an impaired LV diastolic filling that may, in part, be related to the reduction in venous return arising from blood retention within the contracting muscles. Moreover, an abnormal increase in LV filling pressures during IE may also have played a role in reducing LV filling. We observed that, at peak exercise, the E/E′ ratio was higher and the deceleration time was significantly shorter compared to rest values in the IHD group. Both of these findings suggest that IE was responsible for a significant increase in LV filling pressure in this group [32]. These results were also mirrored at the atrial level by the reduction in PALS values compared to rest, indicating an increased LA pressure [33]. Taken together, these data suggest the occurrence of LV diastolic dysfunction during the exercise phase in IHD. The significant correlation that we found between changes in GWW and changes in E/E′ ratio let us hypothesize that the rise in LV filling pressure further worsened the already-impaired LV contractile efficiency. This result seems to comply with those of other recent studies. D’Andrea et al. [17] showed that GWE at rest was closely related to maximal watts reached as well as to LV E/E′. It should be noted that, in the present study, we documented a significant increase in BP in the IHD group, while only modest changes in BP occurred in HC. Considering that, in this research, IE involved large muscle groups, a considerable rise in BP during IE was an expected result. The size of the muscles involved in the IE affects the BP response, and it has been shown that, at the same intensity, isometric bilateral leg extension elicits a greater increase in BP compared to handgrip [34]. It has been hypothesized that the greater the muscle mass involved, the greater the activation of SNA, intramuscular pressure, and vascular occlusion generated [35]. However, the probability of incurring an exaggerated BP rise is also higher in hypertensive than in normotensive subjects [35], and this could explain the different BP response that we observed between IHD and HC. The results of our study perfectly match those of previous research carried out using invasive hemodynamic monitoring, in which two different patterns of hemodynamic responses during IE were described [20,36]. The first pattern, like that which we encountered in HC, was characterized by a modest rise in systolic BP coupled with a rise in cardiac index, a significant increase in HR, and no changes in systemic vascular resistance. A second pattern was instead characterized by a conspicuous rise in systolic BP, coupled with little or no change in cardiac index and a small increase in HR. This latter pattern, which was close to what we found in IHD in the present study, has typically been observed in subjects with a certain degree of LV dysfunction, including hypertensive subjects with LV hypertrophy as well as patients with IHD [22,37]. Our results suggest that, for hypertensive patients with IHD, performing bilateral isometric leg extension at 30% of the MVC is clinically safe, although energetically expensive at the myocardial level. The use of speckle tracking echocardiography with evaluation of the MW technique appears to be useful for monitoring the contractile response of the left ventricle to exercise. Given the potential utility of IE for treating hypertension in these patients, further studies are needed in order to find out the most energetically convenient type of IE for the LV of IHD patients. In particular, changes in MW during isometric leg extension performed at lower intensities or during handgrip performance should be assessed.

### 4.3. Limitations

The results of the present research were obtained in a small group of subjects, and further confirmations in larger studies are needed. The design of the present study, and, in particular, the lack of a group that did not perform the exercise, allowed us to reach only a relatively low level of evidence. Finally, because of the small sample size, there is a certain degree of overlap in the SDs of different variables, which makes the data interpretation difficult. Taken together, these limitations require us to be cautious when interpreting the results and drawing conclusions. The study tested the hemodynamic response to an isometric exercise that involved bilateral leg extension and a load corresponding to 30% of the MVC. In the present study, the intensity of the isometric load was set according to data reported by the literature. In most studies, the intensity chosen for both handgrip and isometric bilateral leg extension, in hypertensive subjects, was between 20 and 50% of the MVC [38]. Clearly, we cannot rule out that IE involving different muscle groups or performed at different intensities could evoke a different myocardial response. For example, it has been demonstrated that BP increases are proportional to the amount of skeletal muscle that is contracting and that handgrip requires a lower BP increase than leg extension [35]. In this study, very small proportions of subjects in both groups were female; therefore, we think that our results cannot be generalized to the female gender. Future studies should consider a design that accounts for gender differences. In the IHD group, we enrolled only patients with underlying coronary diseases. However, we cannot rule out the concomitant, non-recognized presence of a phenotype negative but genotype positive hypertrophic cardiomyopathy [39] that could have potentially affected the results obtained in that group of patients. We did not test the reproducibility of the IE test, and this could have generated errors in the identification of the MVC and, correspondingly, in the load used for the exercise. Although, in this study, we did not test the reproducibility of MW measurements, we utilized data from our previous research in which we assessed the reproducibility of all echocardiography measurements and in which we used the same echocardiographic device as in the present study [40]. Although all patients included in the IHD group carried the same diagnosis, there were also some important differences between them: four out of twenty had not had a myocardial infarction, and seven had EF less than 50%. Finally, differences in pharmacological therapy may have influenced the hemodynamic responses presented by these patients. We believe that these differences increased the heterogeneity of the sample, limiting the generalizability of the results of this study. Further studies are needed to better clarify the variables that determine the hemodynamic response in these patients.

## 5. Conclusions

In hypertensive patients with underlying IHD, the cardiac response occurring during bilateral isometric knee extension was characterized by a dysfunctional increase in contractility that, together with a blunted chronotropic response, prevented the increase in cardiac output. Further studies are needed for identifying the best-tolerated “dose” of IE in the context of cardiac rehabilitation.

## Figures and Tables

**Figure 1 jcm-13-05955-f001:**
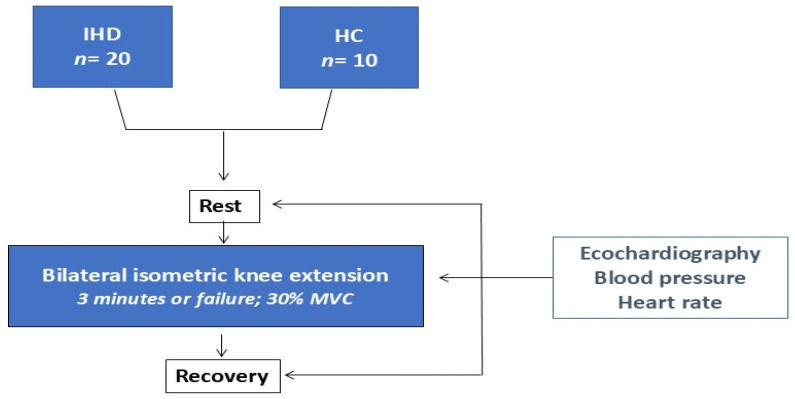
Study flowchart. IHD: Ischemic Heart Disease; HC: Healthy Controls; MVC: Maximal Voluntary Contraction.

**Figure 2 jcm-13-05955-f002:**
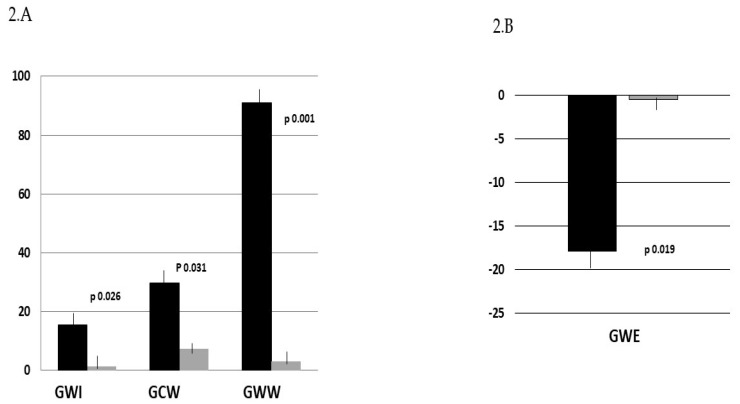
Percentage changes in GWI, GWC, GWW (**2.A**), and GWE (**2.B**) during isometric knee extension (peak exercise vs. rest) in the IHD (black bars) and HC (gray bars) groups. IHD = ischemic heart disease; HC = healthy controls; GWI = global work index; GCW = global constructive work; GWW = global wasted work; GWE = global work efficiency.

**Figure 3 jcm-13-05955-f003:**
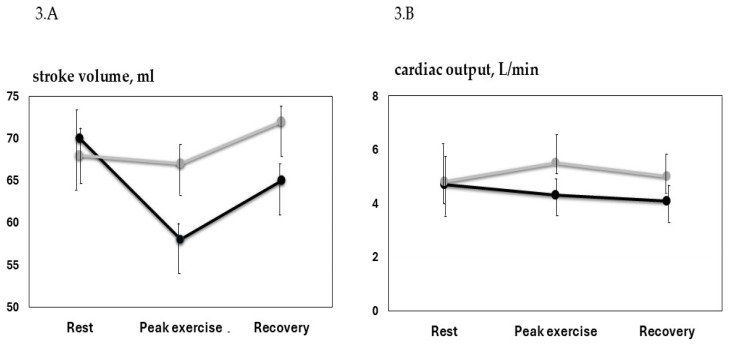
Changes in stroke volume (**3.A**) and cardiac output (**3.B**) during isometric exercise in the IHD (black line) and HC (gray line) groups.

**Table 1 jcm-13-05955-t001:** Anthropometric and clinical characteristics of patients.

	IHD (*n* = 20)	HC (*n* = 10)
Age, years	63.4 ± 7.5	61.8 ± 4.8
BMI, kg/m^2^	27.5 ± 6.3	26.7 ± 7.1
Male/female	16/4	7/3
Previous PCI/CABG	17/8	-
EF, (%)	50.3 ± 7.4 ^†^	58.4 ± 3.6
NT-pro BNP	95.0 ± 59.3	22
Comorbidities		
Carotid artery disease, *n* (%)	8 (40)	-
Hypertension, *n* (%)	20/(100)	-
Diabetes, *n* (%)	5 (25)	-
Hypercholesterolemia, *n* (%)	18 (90)	4 (40)
Previous Smoke habit, *n* (%)	13 (65)	3 (30)
Treatment		
Anti-platelet agents, *n* (%)	20 (100)	
ACE-Is/ARBs, *n* (%)	18 (90)	-
Betablockers, *n* (%)	20 (100)	-
Diuretics, *n* (%)	8 (40)	-
Ranolazine, *n* (%)	4 (20)	
Ivabradine, *n* (%)	1 (5)	
Statins, *n* (%)	20 (100)	3 (30)
Ezetimibe, *n* (%)	12 (60)	1 (10)

IHD = ischemic heart disease; HC = healthy controls; BMI = body mass index; PCI = percutaneous coronary intervention; CABG = coronary artery bypass grafting; EF = ejection fraction; NT-pro BNP = N-terminal pro b-type natriuretic peptide; ACE-Is = angiotensin-converting enzyme inhibitors; ARBs = angiotensin receptor blockers; ^†^ between-groups *p* < 0.005.

**Table 2 jcm-13-05955-t002:** Changes in hemodynamic and echocardiography parameters during isometric exercise in the IHD and HC groups.

	IHD (*n* = 20)			HC (*n* = 10)		
	Rest	Peak	Recovery	Rest	Peak	Recovery
HR, b/min	68.7 ± 10.2	77.1 ± 14.6	63.1 ± 7.7	71.4 ± 12.0	91.1 ± 9.3 *^,†^	75.1 ± 11.0
SBP, mmHg	124.6 ± 15.8	162.2 ± 34.6 *^,†^	121.5 ± 14.6	124.1 ± 13	132.5 ± 11.2	120.7 ± 16.5
DBP, mmHg	77.8 ± 9.3	97.2 ± 20.2 *^,†^	76.5 ± 6.0	75.0 ± 11.3	82.4 ± 7.6	76.3 ± 12.9
DP	8560.0 ± 952.7	12,505.6 ± 883.1	7666.6 ± 874.5	8860.7 ± 652.8	12,070.7 ± 1023.2	9064.5 ± 789.5
Echocardiography						
LVEDV, mL	140.4 ± 33.4	122.3 ± 35.6 *^,†^	137.1 ± 35.2	118.9 ± 18.2	119.4 ± 15.3	119.6 ± 16.1
LVESV, mL	70.5 ± 11.7	67.2 ± 16.1	72.2 ± 14.5	50.2 ± 7.9	49.3 ± 7.4	47.4 ± 6.5
LVEF, %	50.3 ± 7.4	49.7 ± 7.2	48.3 ± 5.6	58.4 ± 3.6	59.3 ± 3.2	59.5 ± 2.2
LVGLS, %	−11.4 ± 3.8	−7.9 ± 2.7 *^,†^	−12.2 ± 3.8	−18.4 ± 2.4	−18.6 ± 2.5	−20.1 ± 2.6
GWI, %	1050.5 ± 412.6	1212.7 ± 356.4 *^,†^	1145.6 ± 521.8	2033.0 ± 108.5	2025.7 ± 130.2	1573.1 ± 413.3
GCW, %	1501.3 ± 491.8	1958.1 ± 328.3 *^,†^	1391.9 ± 469.0	2231.3 ± 85.8	2114.3 ± 126.7	2085.0 ± 348.6
GWW, %	289.7 ± 127.7	554.9 ± 304.7 *^,†^	218.8 ± 128.7	52.7 ± 5.2	54.4 ± 3.1	58.5 ± 156.5
GWE, %	80.7 ± 11.7	66.4 ± 8.1 *^,†^	82.3 ± 11.2	96.0 ± 2.9	95.3 ± 3.6	95.2 ± 9.6
DT, ms	248.7 ± 60.4	220.3 ± 57.6 *^,†^	254.6 ± 56.3	194.8 ± 38.4	192.1 ± 53.6	192.4 ± 53.2
E, cm/s	48.1 ± 9.0	60.4 ± 15.2	48.5 ± 13.1	55.7 ± 13.4	55.4 ± 17.0	64.3 ± 12.9
A, cm/s	66.4 ± 16.1	77.5 ± 23.2	63.0 ± 15.2	48.6 ± 9.3	54.8 ± 9.8	51.2 ± 11.3
E/A ratio	0.75 ± 0.16	0.84 ± 15.2	0.78 ± 0.2	1.2 ± 0.3	1.5 ± 0.4	1.3 ± 0.4
E′, cm/s	7.4 ± 1.5	5.4 ± 4.4 *^,†^	6.2 ± 3.1	11.7 ± 1.6	11.5 ± 2.8	11.9 ± 2.3
E/E′ ratio	6.7 ± 1.9	10.4 ± 7.4 *^,†^	7.4 ± 2.4	5.2 ± 0.8	5.0 ± 1.1	6.4 ± 2.4
TRV, m/s	1.7 ± 0.4	1.7 ± 0.5	1.9 ± 0.5	1.7 ± 0.4	1.6 ± 0.4	1.7 ± 0.4
PALS, %	19.7 ± 8.5	16.8 ± 6.7	22.2 ± 8.4	22.6 ± 4.7	22.4 ± 8.1	23.7 ± 9.4
PACS, %	−8.9 ± 3.5	−11.7 ± 6.7	−12.3 ± 8.0	−12.6 ± 4.5	−12.5 ± 7.5	−13.4 ± 6.1
LAVI, mL/m^2^	25.6 ± 7.2	21.3 ± 7.8	22.9 ± 7.2	19.4 ± 3.2	20.25 ± 6.2	18.8 ± 3.5
SV, mL	70.3 ± 14.3	58.5 ± 12.1 *^,†^	64.7 ± 14.1	68.6 ± 11.8	67.1 ± 15.2	72.2 ± 6.1
CO, L/min	4.7 ± 1.1	4.3 ± 1.3 ^†^	4.1 ± 1.1	4.9 ± 0.8	5.5 ± 0.7	5.2 ± 0.4

HR = heart rate; SBP = systolic blood pressure; DBP = diastolic blood pressure; DP = double product; GWI = global work index; GCW = global constructive work; GWW = global waste work, GWE = global work efficiency; DT = deceleration time; TRV = tricuspid regurgitation velocity; PALS = peak atrial longitudinal strain; PACS = peak atrial contraction strain; SV = stroke volume; CO = cardiac output. * *p* < 0.05 vs. rest, ^†^ between groups *p* < 0.005.

## Data Availability

The data presented in this study are available upon request from the corresponding authors.

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
