# Peer review of "Acute Changes in Myocardial Work during Isometric Exercise in Hypertensive Patients with Ischemic Heart Disease: A Case–Control Study"

_jcm, 2024, doi:10.3390/jcm13195955_

Round 1

Reviewer 1 Report

Comments and Suggestions for Authors

Congratulations for your intersting work regarding the effect of IE on hypertensive IHD patients. The manuscript is well-written and organized. I suggest you separate your manuscript in paragraphs (eg. the Introduction and Discussion parts) and make some parts more apparent by using subtitles (eg. the Methods part). Otherwise, the content of your manuscript is comprehensible. I would add as a limitation the fact that your study is a case-control study (observational - relatively low level of evidence). Due to this fact as well as due to the small number of patients enrolled, the generalization of your results should be carefully made.

Author Response

Congratulations for your intersting work regarding the effect of IE on hypertensive IHD patients. The manuscript is well-written and organized. I suggest you separate your manuscript in paragraphs (eg. the Introduction and Discussion parts) and make some parts more apparent by using subtitles (eg. the Methods part). Otherwise, the content of your manuscript is comprehensible.

- Thank you for this comment. We accepted the reviewer suggestion and accordingly, we divided introduction and discussion in sub-paragraphs.

I would addas a limitation the fact that your study is a case-control study (observational – relatively low level of evidence). Due to this fact as well as due to the small number of patients enrolled, the generalization of your results should be carefully made.

- We accepted the reviewer comment and accordingly we changed the initial part of the limitations paragraph as follow:

“Results of the present research have been obtained in a small group of subjects and further confirmations in larger studies are needed. The design of the present study, and in particular the lack of a group that did not perform the exercise, allowed us to reach only a relatively low level of evidence. These limitations require us to be cautious when interpreting the results and drawing conclusions.

Reviewer 2 Report

Comments and Suggestions for Authors

This paper is extremely interesting, concluding how in patients with hypertension and underlying ischemic heart disease, an acute isometric load may cause a great increase in systolic BP and LV filling pressure. The paper is good and I have only one minor comment in order to improve the manuscript .Among acute changes in myocardial work during isometric exercise in hypertensive patients with ischemic heart disease, the authors should also discuss a potential impact of a preexistent cardiomyopathy. In particular the impact of a phenotype negative but genotype positive HCM in twenty stable hypertensive patients should be considered and discussed  (DOI: 10.1016/j.ijcard.2021.10.013) in discussion section. At the same time authors should express how to exclude it since HCM is a relatively common often inherited global heart disease, with complex phenotypic and genetic expression and natural history (DOI: 10.1016/j.jacc.2021.12.002). Please amplify discussion and cite the 2 suggested references

Author Response

This paper is extremely interesting, concluding how in patients with hypertension and underlying ischemic heart disease, an acute isometric load may cause a great increase in systolic BP and LV filling pressure. The paper is good and I have only one minor comment in order to improve the manuscript .Among acute changes in myocardial work during isometric exercise in hypertensive patients with ischemic heart disease, the authors should also discuss a potential impact of a preexistent cardiomyopathy. In particular the impact of a phenotype negative but genotype positive HCM in twenty stable hypertensive patients should be considered and discussed  (DOI: 10.1016/j.ijcard.2021.10.013) in discussion section. At the same time authors should express how to exclude it since HCM is a relatively common often inherited global heart disease, with complex phenotypic and genetic expression and natural history (DOI: 10.1016/j.jacc.2021.12.002). Please amplify discussion and cite the 2 suggested references

Thank you for this comment. We accepted the suggestion of the reviewer and add this point in the discussion (limitations sub-paragraph). We also add a new suggested reference (n 42)

"In the IHD group we enrolled only patients with underlying coronary diseases. However we cannot rule out a concomitant, non-recognized presence of a phenotype negative but genotype positive  hypertrophic cardiomyopathy [42], that could have affected the results obtained in that group of patients".      

Reviewer 3 Report

Comments and Suggestions for Authors

Dear authors,

1.      What was the rationale for conducting the isometric exercise at 30% of maximal voluntary contraction? Were other load settings considered for testing?

2.      Could you elaborate on the reasons for the smaller increase in heart rate among IHD patients? Are there other potential factors (such as the influence of beta-blockers) that might explain this?

3.      Given that most participants were male, it is difficult to generalize the findings to women. Future studies should consider a design that accounts for gender differences.

4.      There needs to be further discussion on the differences in blood pressure responses between isometric leg extension and handgrip exercises. This highlights the physiological variations based on the type of exercise performed.

5.      It would be beneficial to discuss the rationale for using non-invasive echocardiographic techniques for assessing MW, including their advantages and limitations.

6.      There needs to be further discussion on the applicability of isometric exercise in IHD patients. Proposing a framework to better understand the effects of exercise therapy on cardiac function would be valuable.

Author Response

  1. What was the rationale for conducting the isometric exercise at 30% of maximal voluntary contraction? Were other load settings considered for testing?

Thank you for this comment. The intensity of the isometric load was set according to data reported by the literature (many studies used  this intensity for both handgrip and isometric leg extension). In the revised version of the manuscript we explained better this point and we added  a new reference (41)

  1. Could you elaborate on the reasons for the smaller increase in heart rate among IHD patients? Are there other potential factors (such as the influence of beta-blockers) that might explain this?

We accepted the reviewer comment.  In the revised version of the manuscript we explained better this point and we added  a new reference (32):

“An inadequate increase in HR during exercise is often observed in patients with IHD ad is related to myocardial ischemia and autonomic dysfunction [32]. However, it must be said that the systematic use of betablockers may have contributed to generating or amplifying this chronotropic incompetence in our IHD patients”.

  1. Given that most participants were male, it is difficult to generalize the findings to women. Future studies should consider a design that accounts for gender differences.

Thank you for this comment. We added this point in the ì limitations paragraph.

  1. There needs to be further discussion on the differences in blood pressure responses between isometric leg extension and handgrip exercises. This highlights the physiological variations based on the type of exercise performed.

Thank you for this comment. We expanded the discussion as follow:

“Considering that in this research IE involved large muscle groups, a considerable rise in  BP during IE was an expected result The size of the muscles involved in the IE affects the BP response and it has been shown that at the same intensity, isometric bilateral leg extension elicits a greater increase in  BP  compared to handgrip [37]. It has been hypothesized that the greater the muscle mass involved, the greater the activation of SNA, intramuscular pressure, and vascular occlusion generated [38]”.

  1. It would be beneficial to discuss the rationale for using non-invasive echocardiographic techniques for assessing MW, including their advantages and limitations.

Thank you for this comment. In the discussion paragraph of the revised version of the manuscript we expanded the discussion as follow:

Our data suggest that the use of MW indices allows us to describe the left ventricular performance during exercise in more depth than other methods such as GLS or EF. In particular, the assessment of MW, before starting a cardiac rehabilitation program, could help clinicians to identify combinations of loads and exercise types that are associated with less myocardial wasting work and with greater contractile efficiency. Clearly  further investigations are needed in order to confirm our results and to explore potential long term clinical advantages of performing speckle tracking echocardiography during exercise in clinical practice.

  1. There needs to be further discussion on the applicability of isometric exercise in IHD patients. Proposing a framework to better understand the effects of exercise therapy on cardiac function would be valuable.

Thank you for this comment. In the discussion paragraph of the revised version of the manuscript we expanded the discussion as follow

“Our results suggest that, for hypertensive patients with IHD, performing bilateral isometric leg extension at 30% of the MVC is clinically safe although energetically expensive at the myocardial level. The use of speckle tracking echocardiography with evaluation of the MW technique appears to be useful for monitoring the contractile response of the left ventricle to exercise.  Given the potential utility of IE for treating hypertension in these patients, further studies are needed in order to find out  the most energetically convenient type of IE for the LV of IHD patients. In particular, changes in MW during isometric leg extension performed at lower intensities or during handgrip performance should be assessed.